# Deep Neural Network Regression to Assist Non-Invasive Diagnosis of Portal Hypertension

**DOI:** 10.3390/healthcare11182603

**Published:** 2023-09-21

**Authors:** Federico Baldisseri, Andrea Wrona, Danilo Menegatti, Antonio Pietrabissa, Stefano Battilotti, Claudia Califano, Andrea Cristofaro, Paolo Di Giamberardino, Francisco Facchinei, Laura Palagi, Alessandro Giuseppi, Francesco Delli Priscoli

**Affiliations:** Department of Computer, Control and Management Engineering (DIAG), University of Rome “La Sapienza”, Via Ariosto 25, 00185 Rome, Italy; wrona@diag.uniroma1.it (A.W.); menegatti@diag.uniroma1.it (D.M.); pietrabissa@diag.uniroma1.it (A.P.); battilotti@diag.uniroma1.it (S.B.); califano@diag.uniroma1.it (C.C.); cristofaro@diag.uniroma1.it (A.C.); digiamberardino@diag.uniroma1.it (P.D.G.); facchinei@diag.uniroma1.it (F.F.); palagi@diag.uniroma1.it (L.P.); giuseppi@diag.uniroma1.it (A.G.); dellipriscoli@diag.uniroma1.it (F.D.P.)

**Keywords:** portal hypertension, neural networks, regression, artificial intelligence

## Abstract

Portal hypertension is a complex medical condition characterized by elevated blood pressure in the portal venous system. The conventional diagnosis of such disease often involves invasive procedures such as liver biopsy, endoscopy, or imaging techniques with contrast agents, which can be uncomfortable for patients and carry inherent risks. This study presents a deep neural network method in support of the non-invasive diagnosis of portal hypertension in patients with chronic liver diseases. The proposed method utilizes readily available clinical data, thus eliminating the need for invasive procedures. A dataset composed of standard laboratory parameters is used to train and validate the deep neural network regressor. The experimental results exhibit reasonable performance in distinguishing patients with portal hypertension from healthy individuals. Such performances may be improved by using larger datasets of high quality. These findings suggest that deep neural networks can serve as useful auxiliary diagnostic tools, aiding healthcare professionals in making timely and accurate decisions for patients suspected of having portal hypertension.

## 1. Introduction

Portal hypertension (PHT) is a condition often related to the evolution of chronic liver disease, and is characterized by an excessive increase in portal venous pressure. This involves an increase in the hepatic venous pressure gradient (HVPG) between the portal vein and the inferior vena cava above the nominal value of 1–5 mmHg [1]. PHT involves the formation of collateral vessels that divert portal blood to the systemic circulation, bypassing the liver. This condition poses a significant clinical challenge and requires prompt diagnosis and management to prevent severe complications and improve patient outcomes.

There are numerous causes of portal hypertension, including cirrhosis, a chronic liver disease characterized by extensive fibrosis and nodular regeneration. Cirrhosis is the most common etiology, accounting for approximately 90% of portal hypertension cases worldwide [2]. Other causes include hepatic vein thrombosis, schistosomiasis, portal vein thrombosis, and congenital abnormalities [3]. PHT gives rise to a multitude of complications that significantly impact patient health and quality of life. These complications include variceal bleeding, ascites, hepatic encephalopathy, hepatorenal syndrome, and spontaneous bacterial peritonitis [4]. Variceal bleeding, resulting from the development of dilated and fragile veins in the esophagus or stomach, is one of the most severe complications and can be life-threatening if not promptly managed. The most common symptoms of PHT are shown in Figure 1.

Given the critical nature of PHT and its associated complications, both prevention and early and accurate diagnosis are of the utmost importance. Traditionally, the diagnosis of PHT has relied on invasive procedures such as hepatic vein catheterization, which measures the HVPG, or imaging techniques like Doppler ultrasound and computed tomography (CT) scans [6]. These methods provide valuable information about the extent and severity of portal hypertension, as well as associated liver abnormalities.

However, these traditional diagnostic approaches have limitations, including invasiveness, potential complications, and limited availability. Moreover, the interpretation of the obtained results requires specialized expertise. Hence, there is a need for alternative diagnostic methods that are non-invasive, readily accessible, and provide accurate and reliable results.

In recent years, there has been growing interest in the application of machine learning and artificial intelligence techniques, particularly deep neural networks, in the field of medical diagnostics. These techniques have shown promising results in various medical domains, including cancer detection, cardiovascular disease diagnosis, and neuroimaging analysis [7,8,9]. In the context of PHT, the utilization of deep neural network classification models may offer a novel and valuable approach to improve diagnostic accuracy and aid in the timely management of this condition.

This paper aims to explore the potential of a neural network regression model for supporting the non-invasive diagnosis of PHT in patients with chronic liver diseases. By leveraging standard clinical data, such a model has the potential to provide useful diagnostic outcomes, facilitating timely interventions and improving patient care.

The main contributions of this work are listed as follows:A review of the related works in the literature is performed and presented to identify artificial intelligence and machine learning methods that have already been applied in the context of PHT diagnosis.An innovative neural network regression approach is developed for supporting the non-invasive diagnosis of PHT.Simulations are performed on two datasets in order to validate the efficacy of the proposed method.

The rest of the paper is organized as follows: Section 2 reports related works describing the use of artificial intelligence for diagnosing PHT in the literature; Section 3 describes the data preprocessing that was performed on the dataset; Section 4 introduces the neural network architecture for supporting PHT diagnosis through regression, and discusses the experimental results of this study; and Section 5 draws conclusions and suggests future works.

## 2. Related Works

In recent years, deep learning has emerged as a powerful paradigm within the field of artificial intelligence and machine learning [10]. With its ability to automatically learn and extract intricate patterns from vast amounts of data, deep learning has revolutionized various domains, ranging from computer vision [11] and natural language processing [12] to speech recognition [13] and robotics [14]. This transformative approach has garnered significant attention and demonstrated remarkable success across diverse applications.

Deep learning, inspired by the structure and functioning of the human brain’s neural networks [15], represents a subset of machine learning algorithms that learn multiple layers of representations or features from data [16]. It leverages artificial neural networks composed of interconnected layers of artificial neurons, enabling the modeling of complex relationships and capturing intricate hierarchical patterns. This ability to automatically extract relevant features has led to breakthroughs in areas that were previously considered challenging or unsolvable.

One of the key strengths of deep learning lies in its ability to learn directly from raw data, eliminating the need for explicit feature engineering [17]. This characteristic enables deep learning models to automatically discover relevant features and representations, effectively reducing the burden on human experts.

Artificial intelligence can be used to non-invasively diagnose HVPG through clinical tests such as abdominal ultrasound, computed tomography, magnetic resonance imaging, liver elastography, and the measurement of liver volume, increases in which correlate with disease severity. In particular, deep learning algorithms present the best performance in terms of quantifying HVPG non-invasively [18]. In [19], an auto-machine-learning model based on computed tomography achieves 80% accuracy in identifying severe PHT, implying useful support for early diagnosis and prophylaxis of the disease. In [20], a classification is performed that can discern between instances with HVPG values below 10 mmHg, and instances with HVPG values above 10 mmHg, using a set of deep learning methods on a dataset containing demographic, clinical, instrumental, and laboratory data. Specifically, the performance of four algorithms are compared: Naive Bayes, Regression, K-Nearest Neighbors, and Decision Trees. In [21], the segmentation of liver and spleen is performed to calculate their volume from the computed tomography of the volume of portal veins. Specifically, the deep learning algorithm automatically classifies the obtained segments as belonging to the liver, the spleen, or the background of the image, achieving 95% prediction accuracy. In [22] a machine learning approach was used to predict the HVPG, based on the histological features extracted from picrosirius-red-stained liver biopsies. The inclusion of the enhanced liver fibrosis score, platelet count, aspartate aminotransferase, and bilirubin significantly increased the diagnostic precision. However, this strategy was founded on information from an intrusive liver biopsy. In [23], neural networks were employed as a classification prediction model in a different prospective study on patients who underwent liver biopsy and HVPG measurement in order to identify those who had cirrhosis, CSPH, and esophageal varices. However, this method, which relied on a number of common blood indicators, did not surpass liver stiffness assessment as a stand-alone predictor.

In this work, a neural network regression model was implemented for supporting the non-invasive diagnosis of PHT in patients with chronic liver diseases, using the dataset from [24]. This dataset consists of entries coming from multiple clinical centers in different geographical regions, allowing the satisfaction of the principle of robustness of AI, namely the need for training and testing on heterogeneous data [25]. In [24], the authors pose a classification problem, as is conventional in the academic literature, while in this work a regression problem is tackled. Therefore, the goal is to predict a continuous numerical value rather than a discrete class label. This provides clinical personnel with a directly comprehensible prediction about the value of HVPG, which is also inherently explainable.

## 3. Data Preprocessing

The dataset is composed of retrospective entries from 1233 patients with compensated advanced chronic liver disease (cACLD), gathered from eight clinical centers during a study specifically aimed at PHT; 23 features are present, and 5 of them are selected as the most informative for predicting the presence of PHT [24], which is identified by the value of HVPG. The exclusion of the other features is performed by the authors of [24] through recursive feature elimination (RFE), a machine learning technique for feature selection: it is used to select the most important features from a given dataset to improve the performance of a predictive model while reducing dimensionality. RFE works by iteratively training a model on subsets of features and ranking them based on their importance. The five features taken into account are as follows:Platelet count (PLT) measured in platelets/µL. Platelet count refers to the measurement of the number of platelets, also known as thrombocytes, in a given volume of blood, which is an important parameter for assessing the body’s ability to form blood clots and maintain proper hemostasis.Total serum bilirubin (BILI) measured in mg/dL. Total serum bilirubin refers to the measurement of the concentration of bilirubin, a yellow pigment produced during the breakdown of red blood cells, in the blood. It is an essential component of liver function tests. Elevated levels of total serum bilirubin may indicate impaired liver function or an underlying medical condition.Gamma-glutamyltransferase (GGT) measured in U/L. Gamma-glutamyltransferase is an enzyme found in various tissues, but it is predominantly present in the liver. GGT plays a crucial role in the metabolism of glutathione, a potent antioxidant involved in the detoxification of harmful substances in the body.Activated partial thromboplastin time (aPTT) measured in seconds. Activated partial thromboplastin time is a laboratory test that measures the time it takes for blood to clot and is used to evaluate the intrinsic pathway of the coagulation cascade and monitor the effectiveness of heparin therapy.Cholinesterase (CHE) measured in U/L. Cholinesterase refers to a group of enzymes that are responsible for breaking down the neurotransmitter acetylcholine in the body, regulating its levels and terminating its action.

The label is provided by a sixth feature, namely hepatic venous pressure gradient (HVPG): a person is considered healthy if HVPG ≤ 10 mmHg, whereas if 10≤ HVPG ≤16 mmHg the person has significant PHT, and if HVPG ≥ 16 mmHg the person has severe PHT.

The other features in the dataset are center name, center-ID, age, sex, disease activity, etiology (ALD, viral infections, NASH), etiological therapy, strict compensation, date of HVPG measurement, date of laboratory tests, number of days between HVPG measurement and laboratory tests, HVPG, Liver Stiffness Measurement, interquartile range of Liver Stiffness Measurement, International Normalized Ratio, score of Model for End-Stage Liver Disease, Child–Pugh score, and comments, if any.

An example of some lines from the considered subsection of the dataset is reported in Table 1.

The deep neural network regressor is trained on the dataset subsection coming from a clinical center (Vienna) that presents all non-null values, and tested on the data of another clinical center (Madrid).

Figure 2 shows the class occurrences in the Vienna cohort: note that such datasets tend to be quite balanced.

Figure 3 shows a preliminary analysis that was performed in order to visualize potential correlations between the distribution of PHT classes (healthy person, patient with significant PHT, or patient with severe PHT), and the distributions of the clinical features that are used to perform regression (PHT, BILI, GGT, aPTT, and CHE).

In particular, it was observed that

Lower values of PLT are associated with more frequent occurrences of PHT, i.e., inverse proportionality between PLT and HVPG was presumed;Higher values of BILI are associated with more frequent occurrences of PHT, i.e., direct proportionality between BILI and HVPG was presumed;Higher values of GGT are associated with more frequent occurrences of PHT, i.e., direct proportionality between GGT and HVPG was presumed;Higher values of aPTT are associated with more frequent occurrences of PHT, i.e., direct proportionality between aPTT and HVPG was presumed;Lower values of CHE are associated with more frequent occurrences of PHT, i.e., inverse proportionality between CHE and HVPG was presumed.

The Pearson correlation coefficient ρ [26] is used to further describe the relationships between HVPG and the other clinical features. It is defined as follows:(1)$$\rho =\frac{\sum_{i=1}^{n}\left(x_{i}-\bar{x}\right)\left(y_{i}-\bar{y}\right)}{\sqrt{\sum_{i=1}^{n}{\left(x_{i}-\bar{x}\right)}^{2}}\sqrt{\sum_{i=1}^{n}{\left(y_{i}-\bar{y}\right)}^{2}}},$$
where $x_{i}$ and $y_{i}$ represent the individual data points of two features, $\bar{x}$ and $\bar{y}$ represent the mean values of the two features, and the denominator is the product of the standard deviations of the two features. Figure 4 reports the Pearson correlation coefficients.

The numerical values of the dataset were rescaled between 0 and 1, i.e., MinMax normalization was applied. Each data point $x^{i}$ was transformed using the following formula:(2)$$x_{norm}^{i}=\frac{x^{i}-x_{min}}{x_{max}-x_{min}},$$
where $x^{i}$ is the considered data point, and $x_{min}$ and $x_{max}$ represent the minimum and maximum data points in the dataset, respectively.

Normalization was performed in order to improve the performance of the neural network regressor. As a matter of fact, normalization is particularly useful when dealing with datasets that contain features with different scales or ranges. By normalizing the values, it brings them into a consistent scale, which can improve the performance of machine learning algorithms that rely on distance calculations or require features to be on a similar scale.

The subsets of some of the clinical centers presented some missing values. Such values were replaced through K-means [27], a clustering algorithm that aims to group data points into distinct clusters based on their similarities. It operates by iteratively assigning data points to the nearest centroid and updating the centroids based on the assigned points. The algorithm seeks to minimize the total within-cluster variance, optimizing the compactness of each cluster while maximizing the separation between clusters.

The K-means algorithm can be described as in Algorithm 1 [27].
**Algorithm 1** K-means algorithm  1:**Initialize** the number of clusters (*k*) and maximum iterations (*max_iterations*)  2:Randomly initialize the centroids for each cluster  3:**while** not converged and iterations < *max_iterations*
**do**  4:    **for** each data point **do**  5:      **for** each centroid **do**  6:         Compute the distance to the centroid  7:      **end for**  8:      Assign the data point to the cluster with the nearest centroid  9:    **end for**10:    **for** each cluster **do**11:      Calculate the new centroid as the mean of all data points assigned to that cluster12:    **end for**13:**end while**14:Output the final clusters

In the current application, when the algorithm encounters a missing value for a certain feature, it replaces the value with the centroid of the cluster, which is identified by the values of the other features. The objective of this data preprocessing technique is to enhance generalization and performance: neural networks learn from patterns and correlations in the data. If missing values are not handled properly, they can disrupt such patterns and hinder the network’s ability to generalize well to unseen data. By replacing NaNs with sensible values, the network is enabled to learn from complete and consistent data, leading to improved generalization and performance.

## 4. Deep Neural Network Regression

A deep neural network was trained to perform regression on HVPG based on the knowledge of the other features: PLT, BILI, GGT, aPTT and CHE. The characteristics of the network, implemented in Google Colab using Tensorflow, are illustrated in the following:The architecture is composed of one input layer with five neurons, two hidden layers with 32 and 16 dimensions, respectively, and one output layer with one neuron, with a rectified linear unit activation function, which is used due to its ability to handle non-linear relationships [28]. Figure 5 depicts the architecture of the implemented network.Dropout is applied, which is a regularization technique that mitigates overfitting (i.e., a condition in which the model becomes excessively specialized to the training data, to the extent that it performs poorly on new, unseen data) and improves generalization performance. It randomly sets a fraction of the output values of neurons to zero during each training iteration. Such a process encourages the network to learn robust and less reliant representations by preventing co-adaptation among neurons.The mean square error is used as the loss function, measuring the average squared difference between the predicted values and the actual values of the target variable. Such a loss function calculates the squared difference between each predicted value $y_{\mathrm{pred}}^{\left(i\right)}$ and the corresponding true value $y_{\mathrm{true}}^{\left(i\right)}$, sums up these squared differences across all data points, and then takes the average. The resulting value represents the average magnitude of the errors or residuals in the predictions. It is given by the following formula:
(3)$$MSE=\frac{1}{n}\sum_{i=1}^{n}{\left(y_{\mathrm{true}}^{\left(i\right)}-y_{\mathrm{pred}}^{\left(i\right)}\right)}^{2}.$$It is used as an Adam optimizer, which dynamically adjusts the learning rate based on the first-order and second-order moments of the gradients.

Figure 6 plots the evolution of the loss function for both training and testing.

Figure 7 shows the performance of the network trained on the Vienna subset, tested on a random 10% fraction of the data. This plot depicts the comparison of the predicted values of HVPG with the true values of the test set. The data points are reasonably close to the diagonal, representing the optimal case in which the prediction is equal to the ground truth. The mean absolute error, being expressed by the following formula,
(4)$$MAE=\frac{1}{n}\sum_{i=1}^{n}\left|y_{\mathrm{true}}^{\left(i\right)}-y_{\mathrm{pred}}^{\left(i\right)}\right|,$$
takes the value of 2.7, and the standard deviation (an indication of how spread out the values are from the mean value) is 2.5.

The performances of the trained neural network regressor model were additionally tested on another test set, namely the subdataset gathered in another clinical center (Madrid). Figure 8 shows the loss function and performance of the network in this second case.

Table 2 summarizes the performance of the network tested on the two subdatasets.

Note that the performances of the proposed model were not assessed using the metrics conventionally used for classification problems, as in [24], such as accuracy, since the goal is to predict a continuous numerical value rather than a discrete class label.

## 5. Conclusions

The study presented in this paper explores the use of a deep neural network regression model for supporting the non-invasive diagnosis of portal hypertension (PHT).

The experimental results show reasonable performance in distinguishing patients with PHT from healthy individuals. The model utilizes five clinical features (platelet count, total serum bilirubin, gamma-glutamyltransferase, activated partial thromboplastin time, and cholinesterase) to predict the hepatic venous pressure gradient (HVPG), which is used as a measure of PHT. The model’s performance suggests that deep neural networks can serve as useful auxiliary diagnostic tools in the non-invasive diagnosis of PHT, because of their ability to detect complex, multifaceted and non-linear relations within data by automatically learning hierarchical representations that capture intricate and abstract patterns.

In the following, we report the main limitations of this work, underscoring areas where future research can enhance the robustness and applicability of the proposed method.

Limited Dataset Quality and Size: One of the foremost limitations of our study is the quality and size of the dataset used. The dataset remains relatively small and noisy in comparison to the complexity of the problem. Moreover, the clinical features that are used to solve the regression problem are those that are currently considered to be the most significant for diagnosing portal hypertension and estimating the value of the hepatic venous pressure gradient. A dataset could be of higher quality if it includes more clinical features in addition to the current ones. The inclusion of more extensive and higher quality data would undoubtedly contribute to a more accurate performance.Data Imbalance: An imbalanced class distribution can lead to biased model predictions. A larger and more balanced dataset would help mitigate this limitation and improve the model’s generalization to a broader range of patient profiles.Limitations Related to Prospective Study Design: A longitudinal study with prospective data collection through monitoring patients over an extended period could provide more insights into the predictive capabilities of the proposed method over time.

It is worth noting that the performance of the regression model could be further improved with larger datasets of high quality. As a matter of fact, this study is a tentative elaboration in the framework of the CADUCEO project [29], during which custom datasets will be gathered and used for further model training and testing.

Additionally, future research could explore the incorporation of other clinical variables and imaging data to enhance the accuracy and reliability of the diagnostic predictions.

In conclusion, this study demonstrates the potential of deep neural networks as valuable tools for supporting the non-invasive diagnosis of portal hypertension. The use of readily available clinical data and the elimination of invasive procedures can aid healthcare professionals in making timely and accurate decisions for patients suspected of having PHT.

## Figures and Tables

**Figure 1 healthcare-11-02603-f001:**
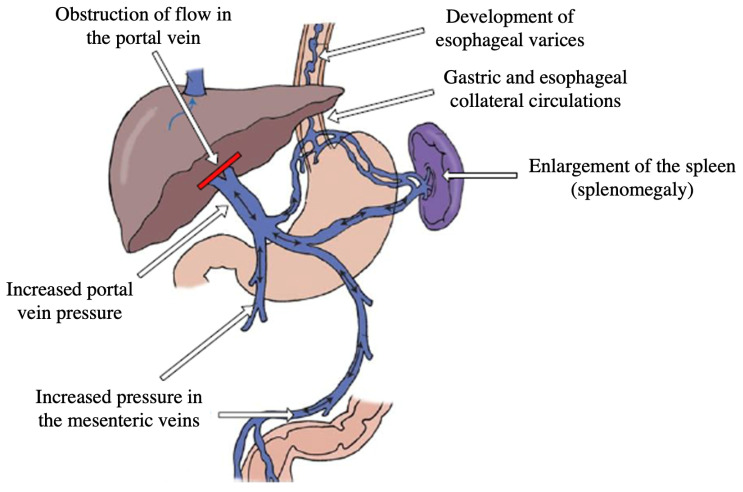
Symptomatology of portal hypertension [5].

**Figure 2 healthcare-11-02603-f002:**
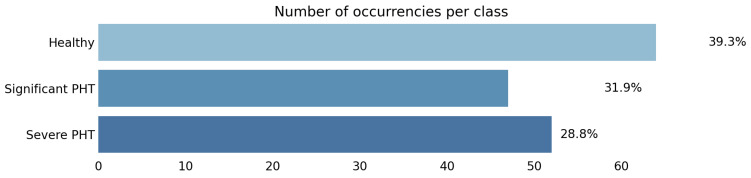
Class occurrences in the dataset.

**Figure 3 healthcare-11-02603-f003:**
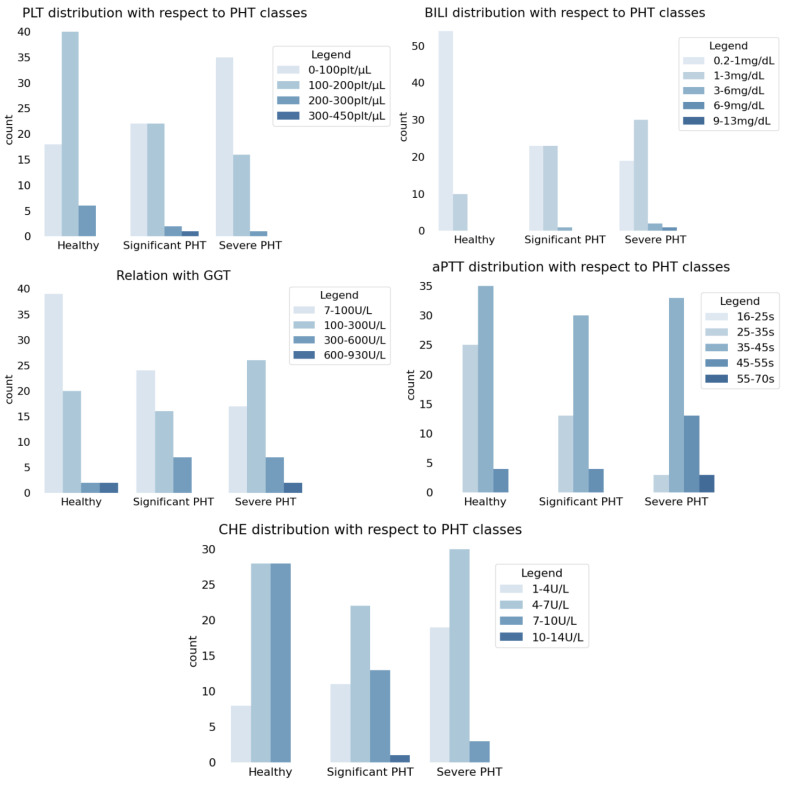
Comparison of the distributions of PHT classes with the distributions of PLT, BILI, GGT, aPTT and CHE.

**Figure 4 healthcare-11-02603-f004:**
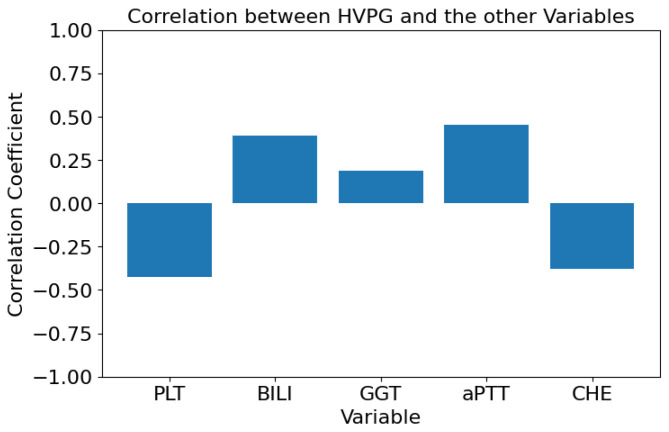
Pearson correlation coefficients between HVPG and the other clinical features.

**Figure 5 healthcare-11-02603-f005:**
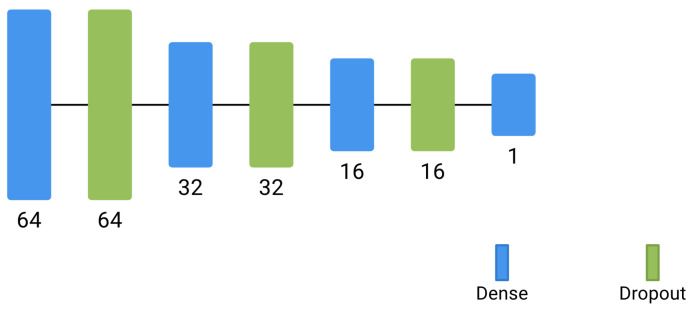
Architecture of the implemented network.

**Figure 6 healthcare-11-02603-f006:**
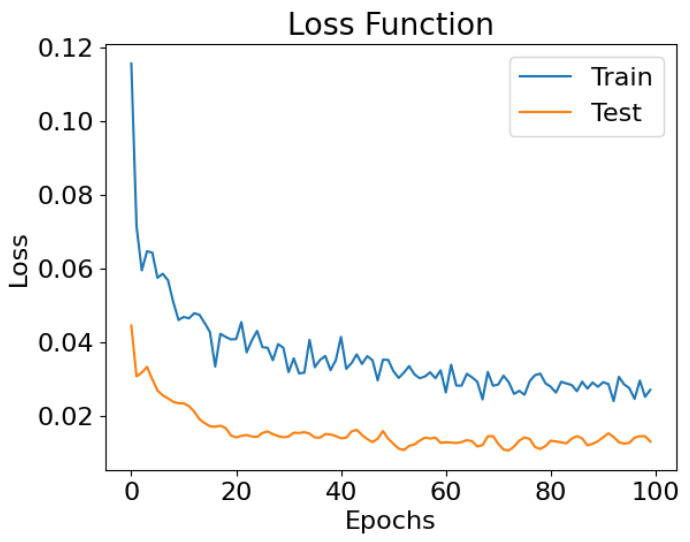
Loss function evolution for training and testing over 100 epochs.

**Figure 7 healthcare-11-02603-f007:**
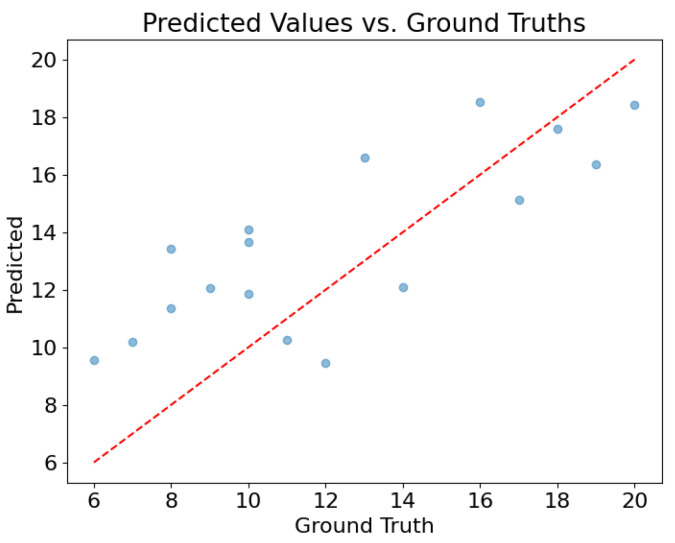
Performance of the network trained and tested on the Vienna subdataset.

**Figure 8 healthcare-11-02603-f008:**
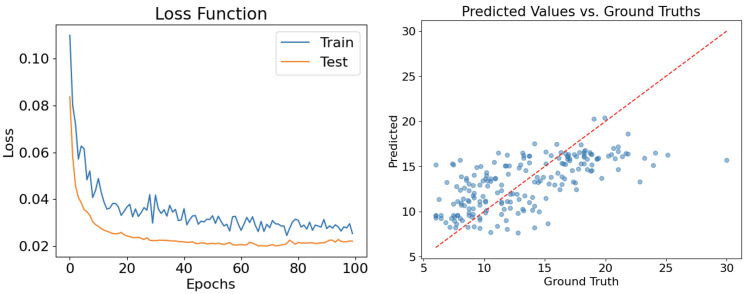
Loss function evolution for both training and testing over 100 epochs, and the performance of the network trained on the Vienna subdataset and tested on the Madrid subdataset.

**Table 1 healthcare-11-02603-t001:** Example entries of the dataset.

Patient ID	PLT (G/L)	BILI (mg/dL)	GGT (U/L)	aPTT (s)	CHE (kU/L)
1	108.0	0.47	19.0	38.9	6.28
2	171.0	0.51	67.0	34.9	7.74
3	106.0	0.84	304.0	33.5	7.63
4	131.0	1.03	263.0	41.5	7.70
5	133.0	0.87	81.0	38.7	9.06

**Table 2 healthcare-11-02603-t002:** Performances of the network for each test set.

Test Set	Mean Absolute Error	Standard Deviation
Vienna	2.70	2.65
Madrid	2.88	3.55

## Data Availability

Not applicable.

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
