# Peer review of "Deep Neural Network Regression to Assist Non-Invasive Diagnosis of Portal Hypertension"

_healthcare, 2023, doi:10.3390/healthcare11182603_

Round 1

Reviewer 1 Report

The study "Deep Neural Network Regression to Aid Non-Invasive Diagnosis of Portal Hypertension" is of special interest for AI-based diagnostic science.

To improve the quality of the manuscript, the authors must detail:

1. Present a comparative study of the results achieved against the conventional diagnosis of said disease. It is important to present the limitations of each type of comparative study. Example: accuracy.

2. The manuscript strives to demonstrate the effectiveness of training and validating the deep neural network regressor. But, in the end, they manage to distinguish patients with portal hypertension from healthy individuals. It is important to know more details that can be used by the deep neural network with the suggested larger and high-quality data sets.

3. Clearly explain the exclusion parameters of the 23 characteristics to reach only 5.

4. Is it necessary in statistics to measure the mean square error (MSE) or mean square deviation (MSD)?

5. Update the references, because several are over 20 years old.

Please review scientific grammar.

Reviewer 2 Report

This paper aims to evaluate the use of deep neural network regression (DNNR) to assist non-invasive diagnosis of portal hypertension (PHT).

The article is important as it shows the potential of DNNR for a non-invasive diagnosis of PHT.

Overall, the work is interesting and well written.

However, I would suggest to the authors to expand some aspects:

-        The dataset is from the article by Reinis et al. who used machine learning models based on widely available laboratory parameters to develop a non-invasive model to predict the severity of portal hypertension in individuals with compensated cirrhosis, who currently require invasive measurement of hepatic venous pressure gradient.

It would be interesting to highlight the difference between the two articles and the possible advantage of DNNR over the method used by Reinis et al.

 The text makes no mention of the limitations of the study, nor does it provide a presentation of the quality of the dataset (how many and which patients, prospective or retrospective etc.).

 Reference 24 appears in the text after reference 25

Round 2

Reviewer 2 Report

I thank the authors for the clear responses. I have no more questions.